# Is There Risk Compensation among HIV Infected Youth and Adults 15 Years and Older on Antiretroviral Treatment in South Africa? Findings from the 2017 National HIV Prevalence, Incidence, Behaviour and Communication Survey

**DOI:** 10.3390/ijerph19106156

**Published:** 2022-05-18

**Authors:** Nompumelelo Zungu, Musawenkosi Mabaso, Shandir Ramlagan, Leickness Simbayi, Sizulu Moyo, Olive Shisana, Pelagia Murangandi, Ehimario Igumbor, Salome Sigida, Sean Jooste, Edmore Marinda, Kassahun Ayalew, Khangelani Zuma

**Affiliations:** 1Human Sciences Research Council, Cape Town 8000, South Africa; mzungu@hsrc.ac.za (N.Z.); mmabaso@hsrc.ac.za (M.M.); lsimbayi@hsrc.ac.za (L.S.); smoyo@hsrc.ac.za (S.M.); thili.sigida@gmail.com (S.S.); sjooste@hsrc.ac.za (S.J.); emarinda@hsrc.ac.za (E.M.); kzuma@hsrc.ac.za (K.Z.); 2Department of Psychology, University of Pretoria, Pretoria 0028, South Africa; 3Department of Psychiatry and Mental Health, University of Cape Town, Cape Town 7925, South Africa; oshisana@evidencebsol.com; 4School of Public Health and Family Medicine, University of Cape Town, Cape Town 7925, South Africa; 5Evidence Based Solutions, Mandela Rhodes Place, 7th Floor, Corner Wale Street and Burg Street, Cape Town 8000, South Africa; 6Centers for Disease Control and Prevention, Division of Global HIV & TB, Pretoria 0001, South Africa; nny3@cdc.gov (P.M.); ylo8@cdc.gov (K.A.); 7School of Public Health, University of the Western Cape, Bellville 7535, South Africa; ehi.igumbor@gmail.com; 8Department of Epidemiology & Biostatistics, School of Public Health, University of the Witwatersrand, Johannesburg 2193, South Africa

**Keywords:** HIV, risk compensation, ART, sexual behaviour, South Africa

## Abstract

In this paper, risk compensation among individuals on antiretroviral therapy (ART), using the 2017 South African national survey on HIV, is explored. A multi-stage stratified cluster random sampling approach was used to realize 11,130 participants 15 years and older. Logistic regression analysis assessed the association between multiple sexual partners, condom use at last sexual encounter, consistency of condom usage and potential explanatory variables using HIV status and ART exposure as a mediator variable. HIV positive participants who were aware and on ART were less likely to have multiple sexual partners, and less likely not to use a condom at last sex compared to HIV positive participants who were aware but not on ART. The odds of reporting multiple sexual partners were significantly lower among older age groups, females, non-Black Africans, and rural settings, and higher among those with tertiary level education, and risky alcohol users. The odds of no condom use at last sexual encounter were more likely among older age groups, females, other race groups, and less likely among those with secondary level education. The odds of inconsistent condom use were more likely among older age groups, females, and other race groups, and less likely among those with tertiary level education, high risk and hazardous alcohol users. Risk compensation is not apparent among HIV infected adults who are on ART. Risk groups that should receive tailored interventions to reduced risky sexual behaviours were identified.

## 1. Introduction

South Africa had an estimated HIV prevalence of 14% or approximately 7.9 million people living with HIV in 2017, and, among those aged 15 years and older, HIV presence was higher, at 18.8% [1]. The 2017 survey revealed that among all people living with HIV, 62.3% were receiving antiretroviral therapy (ART) [1]. The proportion of people on treatment increased with age from 15 years up and was highest among people aged 50 years or older at 76.7% [1]. Evidence on treatment as prevention [1,2,3], and the adoption of the universal “test and treat“ as an approach to manage HIV, has placed ART scale-up at the centre of achieving an “AIDS-free generation”, not only in South Africa but also globally [4,5,6,7]. Initiation of ART as early as possible after acquiring HIV has been shown to reduce the risk of HIV transmission to uninfected partners by more than 90% [2,3,4,8] and to significantly improve the survival of people living with HIV (PLHIV) [9]. In addition, effective ART can reduce viremia to undetectable levels and prevent onward HIV transmission. 

South Africa has one of the world’s largest ART programmes, with an estimated 70% of PLHIV on ART in 2019 [10]. South Africa adopted a test and treatment model in 2016 [1], and ART scale-up is currently ongoing. The National Department of Health had targeted the initiating of an additional two million people by December 2020 [11]. The enrollment of high numbers of HIV-positive individuals on ART is expected to impact HIV incidence at a population level. However, research suggests that the effects of ART on HIV incidence is dependent on other factors, such as changes in sexual networking dynamics during ART scale-up, adherence to ART, achieving viral suppression and patterns of risky sexual behaviours, including unprotected sex and number of sexual partners [9]. 

There is an ongoing debate on the effect of increased access to ART on risky sexual behavior. At issue is whether increased access to ART, improved viral suppression, and reduced viral transmission may lead HIV-positive individuals to engage in high-risk sexual behaviours they would have otherwise not engaged in without ART treatment, also known as risk compensation [9]. Risk compensation occurs when people engage in higher individual risk behaviours, such as having multiple sexual partners and/or having unprotected sex, due to the increased availability of interventions to prevent and mitigate HIV and other sexually transmitted infections [12,13,14,15,16]. As we increase the availability of ART, it is important to understand the effects on the sexual behaviours of PLHIV, since this has implications for the spread and control of the HIV epidemic. 

Studies on the sexual behaviours of people on ART are inconsistent [17]. However, a significant reduction in risky sexual behaviour among people on ART in sub-Saharan Africa was shown in a meta-analysis [17], though the review could not identify what contributes to the positive behavioural change, and it is unclear whether the observed behavioural changes could be maintained and if it was representative of all sub-Saharan African countries. Other empirical studies have concluded that there was generally limited evidence of risk compensation after ART initiation [9,18,19,20,21,22]. In contrast, others found no difference in sexual risk behaviour, comparing PLHIV on ART and those not on ART [19]. Some studies showed that participants who are not on ART, have more unprotected sexual intercourse than those who are on ART, due to these participants believing that HIV treatment was a sufficient prevention strategy because the risk of transmission is reduced when one is on HIV treatment and viral load has been suppressed [23,24,25]. However, empirical studies have dismissed the increased risk behaviours regarding the perception of reduced HIV transmission risk after ART initiation have largely been dispelled in [9,20,26].

In South Africa, a cohort study found that although unsafe sexual behaviours had decreased among HIV positive individuals after initiation into ART, some proportion did not practice safe sex [18]. It is unclear whether this pattern continues to prevail in South Africa, especially in the context of a massively scaled-up treatment programme and improved survival of patients on ART. This paper explores the question of risk compensation or risky sexual behaviour among individuals on ART in South Africa, using the 2017 South African National HIV Prevalence, Incidence, Behaviour and Communication Survey.

## 2. Materials and Methods

### 2.1. Survey Design and Population

The data used in this paper was obtained from a cross-sectional, population-based household HIV survey conducted in 2017, using a multi-stage stratified cluster random sampling approach described in detail elsewhere [1]. In summary, 1000 small area layers (SALs) were sampled using Statistics South Africa’s 2015 national population sampling frame which consisted of 84,907 SALs [27]. The selection of SALs were stratified by province, locality type (urban, rural formal, and rural informal/tribal areas) and race groups in urban areas, based on the predominant race group in the selected SAL. A total of 15 visiting points (VPs)/households were randomly selected from each of the 1000 SALs, targeting 15,000 VPs. Of these, 12,435 (82.9%) VPs were approached due to lack of access in gated, farm and tribal communities. Among these VPs, 11,776 (94.7%) were valid, and a household response rate of 82.2% was achieved from the valid VPs. This survey included people of all ages living in South Africa. All members in the selected households were invited to participate in the survey [1]. The data was benchmarked to the mid-year estimates for 2017 to generalise the findings to the South African population [27]. 

Informed consent and assent were sought before participants were enrolled in the study. All consenting members of the selected households formed the ultimate sampling unit. A household questionnaire (collected information about the household situation) and three age-appropriate individual questionnaires were used to solicit, among others, sociodemographic information, and sexual history, including HIV related risk behaviours. The questionnaires were administered by field-workers and electronically captured using CSPro software on Mercer tablets. 

### 2.2. Blood Specimen Collection and Processing 

The survey also included collecting a blood specimen for estimating HIV prevalence and ART exposure from consenting participants [1]. Dried blood spot (DBS) samples were collected by finger prick from consenting individuals and were tested for HIV antibodies using an algorithm with three different enzyme immunoassays (EIAs). All samples which were HIV positive during the first two EIAs (Roche Elecys HIV Ag/Ab assay, Roche Diagnostics, Mannheim, Germany and Genescreen Ultra HIV Ag/Ab assay, Bio-Rad Laboratories, Hercules, CA, USA) were subjected to a nucleic acid amplification test (COBAS AmpliPrep/Cobas Taqman HIV-1 Qualitative Test, v2.0, Roche Molecular Systems, Branchburg, NJ, USA) for the final interpretation of test results. Testing for exposure to antiretroviral drugs in HIV-positive specimens was performed using High Performance Liquid Chromatography (HPLC), coupled with Tandem Mass Spectrometry [1]. 

The current study used a sub-sample of data on youth and adults aged 15 years and older who agreed to be tested for HIV, whose blood specimen was screened for the presence of ART, and who responded to the question on awareness of HIV status.

### 2.3. Measures

#### 2.3.1. Primary Outcome and Control Variables

The primary outcome measure consisted of risky sexual behaviours, including condom use at the last sexual encounter, consistent condom use, and the number of sexual partners in the past 12 months. Condom use at the last sexual encounter was based on the question: “Did you use a condom at last sexual encounter with the most recent person you had sex with?” Consistent condom use was based on the question: “How often do you use a condom with your (1) most recent sexual partner (2) second most recent sexual partner and (3) and third sexual partner? Responses were combined into a composite variable and dichotomised into a binary outcome with 1 = every time, 0 = almost every time, 0 = sometimes and never = 0. Multiple sexual partnership is based on the question: “Overall how many sexual partners did you have during the past 12 months”? Responses were coded and dichotomised into risky sexual behaviour indicators as follows:Condom use at last sex (No = 1 and Yes = 0)Consistent condom use (No = 1 and Yes = 0)Number of sexual partners in the past 12 months (One partner = 0 and Two or more partners = 1).

The mediator variable used HIV positive individuals who were aware of their HIV positive status and not on ART as a reference category:HIV positive, aware and not ART = 0HIV negative aware = 1HIV positive aware, and on ART = 2.

#### 2.3.2. Explanatory Variables

Descriptive measures included socio-demographic characteristics, such as age group in years (15–19, 20–24, 25–49, 50 years and older), race (Black Africans and other race groups, which included Whites, Coloureds, and Indians/Asians), marital status (married and not married), educational level completed (no education, primary, secondary, and tertiary), employment status (not employed and employed), locality type (urban, rural informal, rural formal), and alcohol use measured using the AUDIT risk score (0 = abstainers; 1–7 = low-risk drinkers; 8–19 = high-risk drinkers; 20+ = hazardous drinking) [28], which has been validated in South Africa [29].

### 2.4. Statistical Analysis

Descriptive analysis summarised the study sample and risky sexual behaviours by socio-demographic and socio-behavioural factors. Chi-square test was used for comparison of categorical variables. Bivariate logistic regression models were used to assess the relationship between condom use at the last sexual encounter, consistent condom use, number of sexual partners in the past 12 months and potential explanatory variables. In addition, statistically significant variables were entered into a multivariate logistic regression analysis to determine factors jointly and independently associated with selected risky sexual behaviour(s). Levels of risky sexual behaviour(s) were compared between HIV positive individuals who were aware of their HIV status, but were not on ART (reference group), and each of the following two groups: (i) HIV negative individuals who were aware of their HIV status, and (ii) HIV positive individuals who were aware of their HIV status and were on ART. Unadjusted and adjusted odds ratio (AOR) with 95% confidence interval (CI) and *p*-value ≤ 0.05 was used to test for statistical significance. The analysis was weighted to account for the complex multilevel unequal sampling probabilities in the survey design. All analyses were carried out using STATA version 15.0 (Stata Corp, College Station, TX, USA).

## 3. Results

### 3.1. Description of the Study Sample 

Of 11,130 participants 15 years and older (*n* = 1343) 13.7% (95% CI: 12.6–15.0) were aware of their HIV positive status and on ART, (*n* = 645) 7.1% (95% CI: 6.3–7.8) were aware of their HIV positive status and not on ART and (*n* = 9142) 79.2% (95% CI: 77.8–80.6) were HIV negative and aware of their HIV status. Table 1 shows sample characteristics. The majority of the weighted population were 25–49 years, female, Black African, never married, had secondary level education, unemployed, resided in urban areas, and were abstainers from alcohol. In addition, most participants reported multiple sexual partners in the last 12 months, no condom use at the last sexual encounter, and no consistent condom use.

Table 2 shows the distribution of risky sexual behaviours by socio-demographic and alcohol use characteristics. Reporting of multiple sexual partners in the past year was significantly higher among those 25–49 years old, males, Black African, never married, having secondary level education, residing in urban areas, and being hazardous alcohol drinkers. Reporting of no condom use at last sexual encounter was significantly higher among those 50 years and older, females, other race groups, married couples, participants with tertiary level education, employed, and participants residing in urban areas. Participants who reported consistently not using a condom were significantly higher among those 25–49 years old, females, never married, or being abstinent.

### 3.2. Risky Sexual Behaviour by HIV Status and ART Exposure

A comparison of HIV risky sexual behaviours, HIV status and ART exposure is shown in Table 3. Overall, HIV positive participants who were aware of their status and on ART were significantly less likely to have multiple sexual partners in the past 12 months than HIV positive individuals who were aware of their status and not on ART [OR = 0.6 (95% CI: 0.4–0.8), *p* = 0.001]. HIV-negative individuals who were aware of their status were significantly more likely not to use a condom during their last sexual encounter than HIV positive participants who were aware of their status and not on ART [OR = 1.4 (95% CI: 1.2–1.7), *p* < 0.001]. HIV positive participants who were aware of their status and on ART were significantly less likely not to use a condom at the time of their last sexual encounter than HIV positive participants who were aware of their status and not on ART [OR = 0.6 (95% CI: 0.5–0.7), *p* < 0.001]. There was no statistically significant association between HIV status, ART exposure and consistent condom use.

### 3.3. Factors Associated with Risky Sexual Behaviour

#### 3.3.1. Bivariate Logistic Regression Models

Table 4 presents unadjusted bivariate logistic regression models for socio-demographic factors and each of three risky sexual behaviour outcomes using HIV status and ART exposure as a design variable. All statistically significant variables were entered into multivariate logistic regression models. The odds of reporting multiple sexual partners in the past 12 months were significantly lower among older age groups than younger age groups, females than males, other race groups than Black Africans, and those residing in rural informal and formal areas than urban areas. The odds of reporting multiple sexual partners in the past 12 months were significantly higher among participants who were never married than married, those who had higher levels of education than no education or those with primary level education, and those who were alcohol users (low risk, high and hazardous drinkers) than abstainers. 

The odds of reporting no condom use during the last sexual encounter were significantly lower among those who were never married, those who had higher levels of education, and those residing in rural informal areas. In addition, the odds of reporting no condom at the time of their last sexual encounter were significantly higher among older age groups, females, other race groups, and the employed. 

The odds of reporting inconsistent condom use were significantly lower among participants who were never married, those who had higher levels of education, and were alcohol users (low risk, high and hazardous drinkers). In addition, the odds of reporting inconsistent condom use were significantly higher among older age groups, females, other race groups, and those residing in rural informal areas.

#### 3.3.2. Multivariate Logistic Regression Models

##### Multiple Sexual Partnerships

Figure 1 presents multivariate logistic regression models of factors associated with reporting multiple sexual partners in the past 12 months. There was no statistically significant association between HIV status, ART exposure and multiple sexual partners in the past 12 months. The odds of reporting multiple sexual partners in the past 12 months were significantly lower among those aged 25–49 years [AOR = 0.69 (95%CI: 0.56–0.84), *p* < 0.001], and those 50 years and older [AOR = 0.37 (95% CI: 0.24–0.57), *p* < 0.001] than youth aged 15–24 years, females than males [AOR = 0.28 (95% CI: 0.23–0.33), *p* < 0.001], other race groups than Black Africans [AOR = 0.48 (95% CI: 0.36–0.63), *p* < 0.001], rural informal areas [AOR = 0.73 (95% CI: 0.60–0.88), *p* = 0.001] and rural formal areas [AOR = 0.47 (95% CI: 0.33–0.66), *p* < 0.001] than urban areas. The odds of reporting multiple sexual partners in the past 12 months were significantly higher among participants who never married than participants who were married [AOR = 2.87 (95% CI: 2.23–3.71), *p* < 0.001], participants with tertiary level education rather than no education/primary level education [AOR = 1.76 (95% CI: 1.21–2.56), *p* = 0.003], low-risk alcohol drinkers [AOR = 1.64 (95% CI: 1.33–2.02), *p* < 0.001], high-risk alcohol drinkers [AOR = 2.96 (95% CI: 2.37–3.70), *p* < 0.001] and hazardous alcohol drinkers [AOR = 4.42 (95% CI: 2.91–6.73), *p* < 0.001] than abstainers.

##### Condom Use at the Last Sexual Encounter

Figure 2 presents multivariate logistic regression models of factors associated with not using a condom at the last sexual encounter. Relative to HIV positive participants not on ART the odds of not using a condom at the time of their last sexual encounter were significantly higher among HIV negative participants who were aware [AOR = 1.32, CI: 1.12–1.56), *p* = 0.001] and significantly lower among HIV positive participants who were on ART [AOR = 0.54 (95 CI: 0.45–0.66), *p* < 0.001]. The odds of reporting no condom use at the time of their last sexual encounter were significantly lower among participants with secondary level education than participants with no education/primary level education [AOR = 0.81 (95% CI: 0.70–0.94), *p* = 0.004]. The odds of reporting no condom use at the time of their last sexual encounter were significantly higher among those aged 25–49 years [AOR = 1.79 (95% CI:1.59–2.02), *p* < 0.001], 50 years and older [AOR = 4.75 (95% CI: 3.93–7.73), *p* < 0.001] than youth 15–24 years, females than males [AOR1.43 = (95% CI: 1.29–1.57), *p* < 0.001], other race groups than Black African [AOR = 2.55 (95% CI: 2.22–2.94), *p* < 0.001]. 

##### Consistent Condom Use

Figure 3 shows multivariate logistic regression models of factors associated with no consistent condom use. There was no statistically significant association between HIV status, ART exposure and no consistent condom use. The odds of reporting no consistent condom were significantly lower among participants with tertiary level education than no education/primary level education [AOR = 0.23 (95% CI: 0.06–0.88), *p* = 0.031], high-risk alcohol drinker [AOR = 0.33 (95% CI: 0.19–0.59), *p* < 0.001] and hazardous alcohol drinkers [AOR = 0.14 (95% CI: 0.06–0.32), *p* < 0.001] than participants who abstained. The odds of reporting no condom use at the time of their last sexual encounter were significantly higher among participants aged 25–49 years [AOR = 2.09 (95% CI: 1.23–3.57), *p* = 0.007], 50 years and older [AOR = 8.17 (95% CI: 1.86–35.84), *p* = 0.005] than youth 15–24 years, females than males [AOR = 5.95 (95% CI: 3.18–11.11), *p* < 0.001], other race groups than Black African [AOR = 4.44 (95% CI: 1.57–12.51), *p* = 0.005] and rural informal areas than urban areas [AOR = 1.68 (95% CI: 0.95–3.00), *p* = 0.076].

## 4. Discussion

The results of this nationally representative population-based study revealed significantly reduced risk behaviours among individuals that were aware of their HIV positive status and also on ART compared to participants who were HIV positive but not on ART. The results do not provide evidence of risk compensation due to exposure to ART as other studies have suggested [19]. Other studies have also reported reductions in risky sexual behaviour after ART initiation [8,17]. Evidence suggests that HIV counselling and support, associated with engagement with healthcare by people on treatment, help these individuals to limit their risk-taking [30]. Literature suggests a complex relationship between the introduction of ART and change in perceptions of HIV risk, and subsequent changes in behaviour. Others suggest that the heterogeneity of published literature reflects different study designs (longitudinal studies, cohorts, cross-sectional surveys), different study populations (heterosexual couples, key populations, drug users) and different socio-cultural contexts [31]. Therefore the current study contributes to the growing body of literature on the sexual behaviour of people on ART in sub-Saharan Africa; especially important, given mixed and contradictory findings on this topic in the continent.

In this study a few variables were used as indicators of risky behaviour, including multiple sexual partners, condom use at the last sexual encounter and consistent condom use. Reporting multiple sexual partners was significantly less likely among older age groups, females, other race groups, and rural settings, and more likely among those who never married, those with tertiary level education, and alcohol users. Multiple sexual partnerships are one of the sexual risk behaviours placing young people, and especially unmarried men, at risk of HIV infection [32,33]. In addition, educational attainment and alcohol consumption have been identified as being associated with multiple sexual partnerships [34,35]. 

Consistent with current findings, other studies also found substantial variations in condom use behaviour and sociodemographic characteristics. For example, patterns of condom use were significantly less likely among older age groups, females, and other race groups [36,37,38,39,40,41,42]. Power dynamics and type of partner play significant roles in age and gender differences in the pattern of condom use [43,44,45,46]. Studies have shown that condom use, and consistent condom use, were less likely among women, especially among females with an older partner [47]. In such relationships, the older partners are more likely to be the decision makers than their younger or same age partners and they are also less likely to use condoms [47]. Similarly, studies from sub-Saharan Africa have shown the relationship between gender and condom use, where men are more likely to report consistent condom use [48,49,50]. The exact nature of these differences has yet to be elucidated through research studies [49]. Others deduce that gender and relationship constructs are associated with condom use patterns, particularly the masculine nature of male gender identity [51]. 

The findings also indicate that lack of education, or low educational attainment, were associated with no condom use at the time of their last sexual encounter and inconsistent condom use. Elsewhere on the continent, evidence suggests that the level of education is a key determinant of condom use [52]. In these studies, a positive association between educational level and condom use was attributed to higher educational attainment increases response to condom promotion. These observations highlight the need for creative strategies to increase the patterns of condom use among those with either no education or low educational attainment. Interventions may include the promotion of strategies that include community mobilisation and involvement of local organisations. 

In addition, the findings revealed that risky alcohol consumption was associated with no condom use at last sexual encounter and inconsistent condom use. These observations highlight the need for public health interventions targeting both alcohol abuse and inconsistent condom use. Promoting consistent condom use as part of HIV risk-reduction interventions targeting high-risk drinkers is needed. Providing condoms in drinking venues could also be one of the important interventions to increase the level of consistent condom use in this population group [53]. 

Some studies have observed the association between ART and risky sexual behaviour after short durations on ART exposure [54,55,56]. However, others suggest the onset of changes in risky sexual behaviour over a much longer period after ART initiation [56]. These findings highlight the complexity of examining the association of ART and risky sexual behaviour and its variations among study participants in different settings. Therefore, it is important to continue to monitor risk reduction practices with the scaling up of the ART programme. 

This study has some limitations. Sexual behaviours were self-reported and may be subject to recall bias and social desirability bias. In addition, the analysis cannot infer causality due to the cross-sectional nature of the study design, and is therefore limited only to assessing the associations between risky sexual behaviours, HIV status and exposure to ART. It is also important to note that the original data were collected nearly five years ago, and therefore the current situation may be different. Nevertheless, this study provides valuable information regarding risk compensation among HIV-positive youth and adults in South Africa. 

## 5. Conclusions

Risk compensation is not apparent among HIV positive adults who are on ART regarding multiple sexual partnerships, condom use during their last sexual encounter and consistent condom use. Instead, our analysis revealed relatively less risky sexual behaviours among HIV positive individuals who were aware of their status and on ART, compared to HIV positive individuals who were aware of their status and not on ART. The study suggests the need to get all PLHIV on ARTs and interventions to reduce risky sexual behaviours in uninfected people to prevent HIV acquisition; with special efforts to reach the elderly, men, those with no education/low educational attainment and high-risk drinkers. Monitoring long-term trends in risky sexual behaviours among PLHIV after ART initiation remains a priority. Future studies should explore the role of type of sexual partnership/relationship and patterns of risky sexual behaviours among those on ART. In addition, the link between HIV positive status, ART and drivers of multiple sexual partnerships related to risk compensation need further research. Finally, more research is needed on the influence of sex partner characteristics (same age, older or younger) in relation to differences in the pattern of condom use among HIV-positive individuals. 

## Figures and Tables

**Figure 1 ijerph-19-06156-f001:**
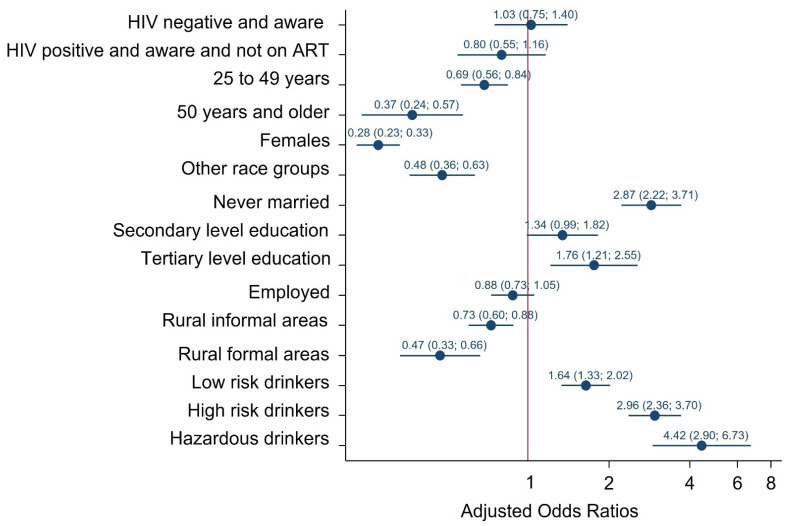
Multivariate logistic model of factors associated with reporting multiple sexual partners in the past year among individuals 15 years older, South Africa 2017 survey.

**Figure 2 ijerph-19-06156-f002:**
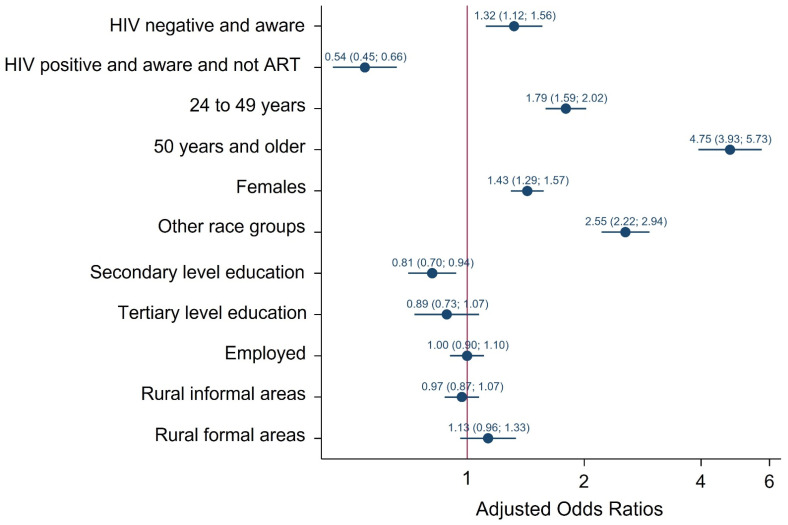
Multivariate logistic model of factors associated with not using a condom during their last sexual encounter among individuals 15 years older, South Africa 2017 survey.

**Figure 3 ijerph-19-06156-f003:**
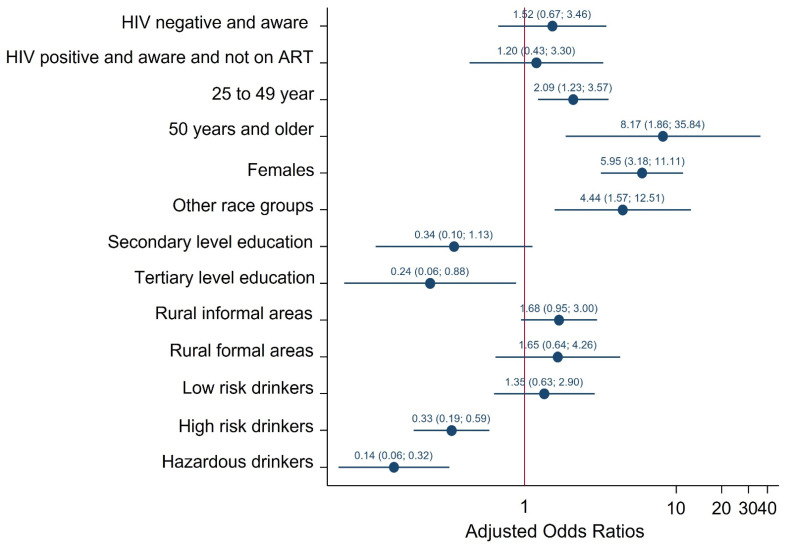
Multivariate logistic model of factors associated with no consistent condom use during their last sexual encounter among individuals 15 years older, South Africa 2017 survey.

**Table 1 ijerph-19-06156-t001:** Participant’s socio-demographic and behavioural characteristics of the study sample, South Africa 2017 survey.

Variables	N	%
Age group in years		
15–24	2354	19.7
25–49	5764	60.4
50+	3023	19.9
Sex		
Male	4046	45.1
Female	7095	54.9
Race groups		
Black African	7856	83.1
Other	3285	16.9
Marital status		
Married	3617	33.6
Never married	6332	66.4
Education level		
No education/primary	1823	15.4
Secondary	5963	68.7
Tertiary	1078	15.9
Employment status		
Not employed	7195	62.7
Employed	3800	37.3
Locality type		
Urban	7211	72.2
Rural informal (tribal areas)	2843	24.2
Rural (farms)	1087	3.6
AUDIT score *		
Abstainers	6944	65.9
Low-risk drinkers (1–7)	2100	22.2
High-risk drinkers (8–19)	928	10.3
Hazardous drinkers (20+)	148	1.6
Numbers of sexual partners in the 12 months		
One partner	6047	88.6
Two or more partners	614	11.4
Condom use at last sex		
No	2360	39.0
Yes	4262	61.0
Consistent condom use		
No	69	1.0
Yes	11,072	99.0

Subtotals do not all equal to the total (N), due to non-response and/or missing data, * alcohol risk score based on a questionnaire for Alcohol Use Disorder Identification Test (AUDIT).

**Table 2 ijerph-19-06156-t002:** Distribution of risky sexual behaviours by socio-demographic and alcohol use characteristics, South Africa 2017 survey.

Variables	Multiple Sexual Partners				No Condom Use at Last Sex				No Consistent Condom Use			
	*n*	%	95% CI	*p*-Value	*n*	%	95% CI	*p*-Value	*n*	%	95% CI	*p*-Value
Age group years												
15–24	1369	18.0	15.2–21.2	<0.001	1357	43.7	40.1–47.4	<0.001	1352	97.3	95.6–98.3	0.014
25–49	4174	11.1	9.7–12.8		4162	61.7	59.5–63.9		4117	98.5	97.7–99.0	
50+	1117	3.4	2.2–5.2		1100	83.1	79.4–86.2		1090			
Sex												
Male	2636	17.7	15.6–20.0	<0.001	2630	58.0	55.4–60.6	<0.001	2601	97.1	95.9–97.9	<0.001
Female	4024	5.5	4.4–6.8		3989	64.0	61.7–66.2		3958	99.8	99.6–99.9	
Race groups												
African	4768	12.6	11.1–14.2	<0.001	4759	56.8	54.8–58.7	<0.001	4714	98.3	97.6–98.8	0.169
Other	1892	5.5	4.1–7.3		1860	83.8	80.9–86.3		1845	99.1	98.0–99.6	
Marital status												
Married	2622	4.1	3.0–5.7	<0.001	2602	82.1	79.3–84.6	<0.001	2574	99.9	99.6–100.0	<0.001
Never married	3767	16.1	14.4–18.0		3746	48.9	46.5–51.3		3718	97.5	96.6–98.3	
Education level												
No education/primary	796	5.3	3.6–7.8	0.005	798	63.3	58.2–68.1	<0.001	788	99.8	99.3–99.9	0.149
Secondary	4049	11.6	10.2–13.3		4024	61.0	58.7–63.3		3992	98.6	97.7–99.2	
Tertiary	769	10.1	7.3–13.9		755	73.0	68.2–77.4		750	98.7	96.7-99.5	
Employment status												
Not employed	3910	11.9	10.4–13.5	0.434	3890	56.7	54.4–59.0	<0.001	3859	98.6	98.0–99.0	0.235
Employed	2681	10.9	9.1–13.1		2663	66.6	63.8–69.4		2634	98.3	96.9–99.0	
Locality type												
Urban	4417	12.3	10.8–14.0	<0.001	4371	62.6	60.3–64.8	<0.001	4326	98.3	97.5–98.8	0.124
Rural informal (tribal areas)	1575	9.4	7.6–11.6		1581	54.5	51.0–58.0		1567	98.8	98.0–99.3	
Rural (farms areas)	668	5.8	3.7–9.0		667	67.6	60.5–74.0		666	99.6	99.0–99.9	
AUDIT score *												
Abstainers	3751	7.8	6.5–9.2	<0.001	3728	59.8	57.4–62.1	0.251	3697	98.9	98.2–99.3	0.001
Low-risk drinkers (1–7)	1490	12.5	9.9–15.8		1473	63.9	60.2–67.5		1459	98.7	96.1–99.6	
High-risk drinkers (8–19)	707	22.0	18.2–26.4		709	60.1	54.9–65.0		697	97.5	95.4-–98.6	
Hazardous drinkers (20+)	112	39.9	26.9–54.5		113	55.0	40.1–69.0		112	90.4	80.4–95.6	

Subtotals do not all equal to the total (*n*) due to non-response and/or missing data, CI—confidence intervals, * alcohol risk score based on a questionnaire for Alcohol Use Disorder Identification Test (AUDIT).

**Table 3 ijerph-19-06156-t003:** Risky sexual behaviour variables by HIV status and ART exposure among individuals 15 years older, South Africa 2017 survey.

Variables	Multiple Sexual Partners				No Condom Use at Last Sex				No Consistent Condom Use			
	OR	95% CI		*p*-Values	OR	95% CI		*p*-Values	OR	95% CI		*p*-Value
HIV positive aware and not on ART	1				1				1			
HIV negative and aware	1.0	0.8	1.3	0.890	1.4	1.2	1.7	<0.001	1.1	0.5	2.2	0.837
HIV positive aware and on ART	0.6	0.4	0.8	0.001	0.6	0.5	0.7	<0.001	1.8	0.7	4.4	0.224

ART—antiretroviral treatment; CI—confidence interval; OR—odd ratio.

**Table 4 ijerph-19-06156-t004:** Unadjusted odds ratios for risky sexual behaviours by socio-demographic factors among individuals 15 years older, South Africa 2017 survey.

	Multiple Sexual Partners				No Condom Use at Last Sex				No Consistent Condom Use			
	OR	95% CI		*p*-Values	OR	95% CI		*p*-Values	OR	95% CI		*p*-Values
Age groups												
15–24	1				1				1			
25–49	0.6	0.5	0.7	<0.001	2.4	2.2	2.6	<0.001	2.2	1.5	3.3	<0.001
50+	0.2	0.1	0.3	<0.001	6.7	5.7	7.8	<0.001	15.8	3.8	65.1	<0.001
Sex												
Male	1				1				1			
Female	0.2	0.2	0.3	<0.001	1.3	1.2	1.5	<0.001	10.8	6.2	18.7	<0.001
Race												
Black African	1				1				1			
Other	0.5	0.4	0.6	<0.001	3.0	2.6	3.4	<0.001	2.1	1.1	4.0	0.018
Marital status												
Married	1				1				1			
Never married	4.3	3.6	5.3	<0.001	0.2	0.2	0.2	<0.001	0.1	0.0	0.2	<0.001
Educational qualification												
None/primary	1				1				1			
Secondary	1.8	1.4	2.3	<0.001	0.6	0.6	0.7	<0.001	0.2	0.1	0.8	0.017
Tertiary	1.8	1.3	2.5	<0.001	0.8	0.7	1.0	0.013	0.2	0.1	0.7	0.016
Employment status												
No	1				1				1			
Yes	0.9	0.8	1.1	0.222	1.43	1.22	1.43	<0.001	1.02	0.69	1.50	0.938
Locality type												
Urban	1				1				1			
Rural informal (tribal areas)	0.8	0.7	0.9	<0.001	0.8	0.7	0.8	<0.001	1.8	1.1	2.8	0.011
Rural (farms)	0.6	0.4	0.7	<0.001	1.1	1.0	1.3	0.133	2.6	1.0	6.3	0.042
AUDIT score *												
Abstainers	1				1				1			
Low risk drinkers (1–7)	1.9	1.6	2.3	<0.001	1.0	0.9	1.2	0.438	1.0	0.5	1.8	0.900
High risk drinkers (8–19)	4.6	3.9	5.5	<0.001	1.0	0.8	1.1	0.516	0.2	0.1	0.3	<0.001
Hazardous drinkers (20+)	7.0	4.9	10.0	<0.001	1.2	0.9	1.7	0.279	0.1	0.0	0.2	<0.001

Subtotals do not all equal to the total (*n*) due to non-response and/or missing data, OR—odds ratio, CI—confidence intervals, * alcohol risk score based on a questionnaire for Alcohol Use Disorder Identification Test (AUDIT).

## Data Availability

The data for this manuscript are openly available on the Human Sciences Research Council institutional repository available at https://repository.hsrc.ac.za/handle/20.500.11910/15468, Archive number: SABSSM 2017 Combined, URI: http://doi.org/10.14749/1585345902.

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
