# Peer review of "Is There Risk Compensation among HIV Infected Youth and Adults 15 Years and Older on Antiretroviral Treatment in South Africa? Findings from the 2017 National HIV Prevalence, Incidence, Behaviour and Communication Survey"

_ijerph, 2022, doi:10.3390/ijerph19106156_

Round 1

Reviewer 1 Report

Thank you for this interesting paper providing some insights into HIV risk behaviour.

The paper is well written and structured overall, and I do have some comments on the sections.

INTRODUCTION

This section provides a helpful background to the topic, risk compensation (or not) in people living with HIV and taking ART. Sources are used appropriately, though I do note that several citations are relatively ‘old’, especially the core papers addressing risk compensation. I would recommend adding more recent studies so support the relevance of your study.

MATERIALS AND METHODS

This section provides useful detail of your methodology, and the source of your data. Your measures and analytical approach seem appropriate, and I note that the original survey was granted ethical approval and there was a suitable consenting process in place at that time.

RESULTS

These are described in depth and provide important insights into risk behaviours associated with HIV. Most importantly, of course, is the finding that suggests in this sample those taking ART show no evidence of risk compensation. The other more general findings are also valuable, especially around those demographic factors that are linked with risky behaviours.

DISCUSSION AND CONCLUSION

You draw on your findings and make suitable recommendations. I would agree that further research is required to confirm the complex interactions between risk behaviour and personal/contextual factors (including the role of ART). Long term monitoring of people taking ART is indeed important to track the impact of support and counselling and the possibility of ‘fatigue’ to risk avoidance over time.

Your limitations are appropriate – I would perhaps also flag that the original data were collected nearly 5 years ago, and therefore the current situation may be different.

REVIEWER RECOMMENDATIONS

  1. In the introduction, gather a couple of more recent studies to highlight issues around risk compensation. A lot has been done, for example, in the context of PrEP.
  2. Note in the limitations that the data are nearly five years old.

Author Response

Hi, 

Reviewer 2 Report

Interesting paper which provides insights on risk behavior, compensation, and socio/anthropologic practices in a high-prevalence demographic of PLWH, which is important to share with scientists and the public.

This is a well-written manuscript with clarity with regards to Introduction, Methods, Results, Discussion.

In Figure 2, change "HIV positive and aware not ART" to "HIV positive and aware and not ART" or something similar (may be missing one word).

In Discussion, you provide explanations for results but would provide more support throughout Discussion via evidence from literature for statements such as, "This may include proportion strategies that include community mobilisation and involvement of local organisations," in order to explain your rationale in an evidence-based manner.

Further expansion/literature review support for condom use variability is needed, in addition to "Power dynamics and type of partner plays a significant role in age and gender differences in the pattern of condom patterns [37-40]."

Here, please put the n value for each (and also it is not clear how many of total individuals surveyed were HIV-positive and HIV-negative so please specify with n value): "

Of 11 141 participants 15 years and older 13.7% (95% CI: 12.6–15.0) were aware of 174 their HIV positive status and on ART, 7.1% (6.3–7.8) were aware of their HIV positive 175 status and not on ART and 79.2% (77.8–80.6) were HIV negative and aware of their HIV 176 status."

Discussion (paragraph 1) should expand and elaborate more upon risk compensation/analysis in the setting of PLWH, with support from literature in the discussion as well.

Author Response

Hi,

Reviewer 3 Report

Authors explored risk compensation or risky sexual behaviour among individuals on ART in South Africa, using the 2017 South African National HIV Prevalence, Incidence, Behaviour and Communication Survey. This paper is good and important to share.

Introduction

Introduction section is quite clear but not effective. Authors mentioned that sexual behaviours of people on ART are inconsistent based on studies done previously. Do not need to explain those results in the introduction. It can enrich your analysis in the discussion section.

Several risk compensation studies have already been done before (9, 18-21). What were the advantages of your research so that this current study was important to do?

Materials and Methods

Since the current study was using secondary data from the 2017 South African national survey on HIV, it was not necessary to explain the blood specimen collection and processing in the Materials and Methods section. Line 121-124 already mentioned which data were used.

Results

Fig 2 and Fig 3.  Should it be Rural informal areas and Rural formal areas? I assumed.  

To make it consistent with previous termination.

Fig 3 missed the x axis label

It is interesting to know duration of ART in people with HIV positive in order to know one of the reason, why they did risky sexual behaviour. Do you have data of ART duration after first initiation?

Minor point:

Write conclusion more simple and do not reiterate your results or the discussion

Author Response

Hi, 

Reviewer 4 Report

Dear Editor,

             The paper is well written and can be published after the mentioned minor changes.

Regards

Author Response

Hi,
